# Initiation of Antenatal Care Among Pregnant Women in Saudi Arabia: An Application of Andersen’s Behavioral Model Using a Cross-Sectional Study

**DOI:** 10.3390/healthcare13192449

**Published:** 2025-09-26

**Authors:** Ajiad Alhazmi, Hassan N. Moafa, Seham A. Habeeb, Reham Bakhsh, Manal Almalki, Jobran Moshi, Ali Saad R. Alsubaie, Hammad Ali Fadlalmola, Mohammed Ali Qassem Ghazwani, Abdulrhman Mohammad Salim

**Affiliations:** 1Department of Public Health, College of Public Health, Imam Abdulrahman Bin Faisal University, Dammam 34212, Saudi Arabia; ahalhazmi@iau.edu.sa (A.A.); asralsubaie@iau.edu.sa (A.S.R.A.); 2Department of Public Health, College of Nursing and Health Sciences, Jazan University, Jazan 82912, Saudi Arabia; moafa@jazanu.edu.sa (H.N.M.); rabakhsh@jazanu.edu.sa (R.B.); almalki@jazanu.edu.sa (M.A.); 3Public Health Department, Jazan Branch, Ministry of Health, Jazan 82611, Saudi Arabia; shabeeballah@moh.gov.sa; 4Department of Medical Laboratory Technology, College of Nursing and Health Science, Jazan University, Jazan 82911, Saudi Arabia; 5Nursing College, Taibah University, Madinah 42351, Saudi Arabia; hafadlelmola@taibahu.edu.sa; 6Health Administration, Primary Health Care, Ministry of Health, Jazan 82721, Saudi Arabia; maghazwani@moh.gov.sa; 7Gazan Health Sector, Infectious Disease Directorate, Ministry of Health, Jazan 82611, Saudi Arabia; absalim@moh.gov.sa

**Keywords:** antenatal care, maternal health, healthcare utilization, Saudi Arabia, Andersen’s behavioral model, pregnancy, barriers to care, healthcare accessibility

## Abstract

**Background/Objectives**: Timely initiation of antenatal care (ANC) services is crucial for ensuring maternal and fetal well-being. Despite the importance of ANC, research regarding its initiation remains limited in the Jazan region of Saudi Arabia, an area with notable adverse birth outcomes. Therefore, this study aimed to assess pregnant women’s initiation of ANC and identify associated factors and significant barriers for timely initiation. **Methods**: A cross-sectional study was conducted among 369 Saudi pregnant women in their third trimester attending ANC clinics in the Jazan region in 2024. A structured questionnaire was used to collect data. Andersen’s behavioral model of healthcare utilization provided the framework for the study. Descriptive statistics, chi-square tests, and binary logistic regression were used to analyze the data. **Results**: The majority of women (78.9%) initiated ANC in the first trimester. Higher maternal education was positively associated with early ANC initiation (aOR = 2.369, 95% CI: 1.154–4.901), whereas higher paternal education was negatively associated with early ANC initiation (aOR = 0.350, 95% CI: 0.175–0.699). When modeled independently, the positive association of higher maternal education was attenuated but was not significant, while the negative association of higher husband’s education remained the same. Those living more than three km from health facilities (aOR = 0.510, 95% CI: 0.276–0.941) and seeking care for reasons other than routine follow-up were less likely to initiate ANC early. Most women received essential services, but only 37.1% had ultrasound tests. **Conclusions**: While ANC initiation in Jazan showed promising trends, factors like geographical accessibility remain a significant barrier. Targeted interventions should address these identified barriers, which fall within predisposing, enabling, need, and external environmental factors. Further investigations of pregnant women’s familial decision-making and low ultrasound test utilization in relation to ANC are recommended.

## 1. Introduction

Antenatal care (ANC) is an essential component of a successful pregnancy. It can be defined as the care provided by skilled healthcare professionals to pregnant women and adolescent girls to ensure the best health conditions for both mothers and babies during pregnancy. The ANC comprehensively addresses broad areas for intervention among pregnant women, including nutritional interventions, maternal and fetal assessments, preventive measures, common physiological symptoms, and health system interventions. These interventions during ANC can save the lives of women and their children since maternal and child health are closely related [1,2,3,4,5,6]. However, pregnancy and childbirth issues are common, especially in low- and middle-income countries. Apart from preventable maternal morbidity, in 2020, it is estimated that there were approximately 800 maternal deaths from preventable causes every day, whereas in 2022, there were 2.3 million newborn deaths [7]. Most of these mortality figures can be prevented by receiving quality maternal healthcare services and the guidance of skilled healthcare professionals [1].

According to the recent guidelines released by the World Health Organization (WHO), the ANC schedule should consist of a minimum of eight visits: one visit initiated during the first trimester, two visits during the second trimester, and five visits during the third trimester. These guidelines emphasize the early initiation of ANC, which is crucial for providing more tailored services, as many recommendations are context-specific [1,8]. Timely initiation helps in the early identification of high-risk pregnancies and promotes patient engagement in service delivery. A study conducted in Tanzania highlighted the role of the late initiation of ANC as a factor that hinders the effective utilization of ANC [9]. Nevertheless, ANC, as a preventive and proactive approach, encompasses several components to promote health during pregnancy; hence, timely initiation is vital for high-quality ANC. Moreover, various studies have highlighted the efficacy of ANC interventions, such as nutritional and preventive interventions, in reducing the prevalence of low birth weight, providing prophylactic agents, and identifying complications during pregnancy [10,11,12,13,14]. However, according to a global systematic analysis conducted from 1990 to 2013, the coverage of early ANC increased from 40.9% in 1990 to 58.6% in 2013; additionally, the study highlighted substantial variations in terms of such figures and regional and income contexts [8]. On the other hand, ANC indicators vary significantly in terms of regional and national perspectives. According to UNICEF, the proportion of women receiving at least four ANC visits is the lowest in Sub-Saharan Africa, at less than 30%, and is the highest in Latin America and the Caribbean, at more than 90% [15]. Several studies have shown that mothers’ educational level, availability, accessibility, previous experience with the health care system, residency, household wealth, and gestational age are examples of determinants of ANC initiation and utilization [16,17].

In Saudi Arabia, the ANC care model is responsive to WHO guidelines, as eight visits are initiated during the first trimester. Furthermore, according to a study conducted at various healthcare institutions in 2017, more than half of the included pregnant women had at least one missed ANC visit, which further delayed their care. The study identified factors associated with ANC attendance, primarily related to the healthcare system and its facilities, as well as the processing of these visits [18]. Later, in 2022, the Saudi Ministry of Health introduced a service titled the “Mother and Child Health Passport” to establish a centralized reference point for the medical history of pregnant women across various healthcare providers during antenatal, intranatal, and postnatal care. This service can function as a case note approach, which is emphasized to foster continual, coordinated, relevant, high-quality, and good pregnancy experiences across the three stages of pregnancy [19]. Another study collected data across 11 centers between 2022 and 2023 and revealed that approximately 15.0% of pregnant women initiated ANC within the first eight weeks; additionally, approximately 65% completed at least eight ANC contacts [20]. Additionally, the study delved deeply into person-related factors, identifying younger ages and fewer pregnancies as negatively associated factors with late ANC initiation, whereas the occurrence of pregnancy-related complications such as abortion or premature birth were positively associated factors.

In the Jazan region of Saudi Arabia, studies investigating the initiation of ANC are limited. Furthermore, a study conducted by Dallak et al. reported a high prevalence of adverse birth outcomes in the Jazan region, with 46.6% of women with at least one adverse birth outcome, predominantly miscarriage (12.9%), followed by preterm and underweight birth (~4.0%) [21]. Additionally, as reported by the same study, the higher prevalence of marriages between cousins/relatives in the Jazan region (49.0%), in addition to higher prevalences of passive smoking (32.0%) and smokeless tobacco use (7.4%), may contribute to the noted figures of adverse birth outcomes. Given this and the recommendation to investigate ANC initiation [8], this study aims to assess the initiation of ANC in the Jazan region among pregnant women and identify associated factors and significant barriers for timely initiation while considering Andersen’s model [22], which proposes factors that influence healthcare use in three broad categories: predisposing, enabling, and need factors. This model posits factors from a comprehensive theoretical view that were not previously investigated or were just partially addressed in other Saudi studies [18,20]. For example, instead of focusing only on clinic-based and limited personal factors, this model allows the inclusion of structural factors in addition to a broad range of predisposing, need, and enabling factors, such as taking into account couples’ ages and educational levels, media exposure, marriage duration, etc.

## 2. Materials and Methods

### 2.1. Study Setting

This cross-sectional study was conducted at various health centers in the Jazan region of Saudi Arabia, a southwestern region characterized by its diverse population and unique cultural and geographical features. It has a total population of 1.6 million inhabitants and 16 major governorates, covering an area of 13,400 km^2^. Jazan city, the region’s capital, is situated on the Red Sea coast and serves as a significant urban center [23]. Healthcare services are provided to all Saudi citizens for free, while non-Saudi citizens receive services through health insurance companies. Most of the healthcare system in the region is operated by the Saudi Ministry of Health, and the region contains approximately 177 primary healthcare centers [24]. These centers provide antenatal care through which high-risk pregnancies can be identified and, if needed, referred to a larger center using the Mother and Child Health Passport [19].

### 2.2. Study Design

A cross-sectional study design was employed to investigate the factors associated with ANC initiation among pregnant women in the Jazan region. Data were collected from 369 pregnant women who attended ANC clinics in 2024. Moreover, this study complies with the Strengthening the Reporting of Observational Studies in Epidemiology guidelines, as shown in Table A1 [25].

### 2.3. Study Population

The study purposively recruited Saudi women from ANC clinics who were aged 18 years and older, were pregnant, and had entered their third trimester. Women who were not Saudi citizens, were under 18 years of age, were not pregnant or were pregnant but not in their third trimester, had pregnancy complications or comorbidities, had psychological problems, or refused to participate were excluded from the study. The exclusion of pregnant women with complications is crucial, especially because we aimed to capture routine ANC initiation rather than as a result of other medical necessity. Therefore, it is important to note that the study captures only pregnant women who attended antenatal care clinics. The sample size was estimated for this cross-sectional study via the following equation: n = z^2^pq/d^2^ [26]. Confidence intervals of 95% and 5% margins of error were utilized. Additionally, we used the finding that 66.0% of people initiated ANC early, based on a study conducted in Riyadh [18]. Finally, a minimum of 345 pregnant women were required to participate in the study.

### 2.4. Study Framework

This study was conducted within the framework of Andersen’s behavioral model of healthcare services utilization. As shown in Figure 1, this model includes multiple factors distributed over three categories. These include the educational levels of mothers and husbands, their ages, the number of children, household decision-makers, and marriage duration as predisposing factors. Need factors include perceived health conditions, such as disease signs and symptoms like fever. Enabling factors include the occupation of the mother and her husband, their income, proximity to the health center, media exposure, and the availability of health centers. This model also emphasizes external environmental factors, such as the availability of certain services within the healthcare system [22]. A recent systematic review by Alkhawaldeh et al. (2023) confirmed the applicability of Andersen’s model in various healthcare contexts, including maternal health services [27].

### 2.5. Data Collection

A structured questionnaire was used to collect participants’ data. Pregnant women were informed regarding study objectives, estimated time for data collection, nature of voluntary participation, and right to refuse or withdraw at any phase of the study without any consequences. Upon obtaining verbal informed consent, trained data collectors conducted face-to-face interviews using the structured data collection tool. Verbal consent ensured inclusivity, minimized potential anxiety among participants about signing official forms, and respected sensitivities in terms of varying levels of literacy. This instrument included three distinct sections. In the first section, we collected data related to sociodemographic characteristics. These characteristics included the ages of the mother and her husband, educational levels, marriage duration in years, employment status, parity, income in Saudi Riyal (SR), distance to the health center in km, media exposure (e.g., cell phone, television, newspapers), reasons for ANC visits, ANC provider, and knowledge regarding the recommended ANC visits. Educational level was collected according to the Saudi educational stage. These stages include illiterate/no formal education and primary, intermediate, secondary, university, and graduate studies. The reasons for ANC visits included periodic follow-up, swelling of hands/face, high blood pressure, high fever, abdominal pain, blurred vision, headache, severe vaginal bleeding, and other factors. ANC providers are categorized into three categories: nearby health centers, governmental health centers, and private health centers. The second section was used to collect the initiation time of ANC by indicating the trimester in which ANC was commenced. In the third section, we asked the participants to specify the services they had received during their previous ANC visits. The list consisted of basic services during ANC, including clinical examinations, blood pressure and weight measurements, blood tests, vaccines, medicines, ultrasound tests, and health education. Moreover, we asked participants to identify their barriers if they had missed at least one ANC visit. The list included the following specific barriers: no ultrasound services, no lab services, no consultant, the physician did not recommend that I have to visit, I feel healthy, and other (with participants having to specify the other barrier). Where needed, electronic health records were used complementarily to collect participants’ data. Finally, participants were grouped in terms of ANC initiation into first-trimester and second-/third-trimester groups in accordance with the WHO schedule of ANC visits [1].

### 2.6. Pilot Study

A pilot study was conducted using 23 participants who were not included in the actual study to ensure that the study tool for data collection was valid and reliable. Their comments were addressed accordingly. Furthermore, we formed a construct named “lack of services or facilities” by grouping the following barriers: no ultrasound services, no laboratory services, and no consultants. More importantly, the measure of sampling adequacy was 0.628, with 67.5% of the variance explained, accompanied by a significant Bartlett test of sphericity < 0.05, confirming the suitability of factor analysis. The factor loadings were calculated as follows: 0.849, 0.889, and 0.716 for no ultrasound, no lab, and no consultants, respectively. Additionally, Cronbach’s alpha was calculated to ensure reliability. The tests indicated that the components meaningfully contributed to the construct, with a Cronbach’s alpha greater than 0.7, as shown in Table A2.

### 2.7. Data Management and Analysis

Certain variables were grouped due to their low frequency, which could significantly affect the statistical tests. For example, upon noticing higher standard or sampling errors (SE), educational levels were grouped into “below university degree” and “university degree or above”. Furthermore, the causes of ANC visits were grouped as follows: periodic follow-up visits, swelling of hands, and others (high blood pressure, blurred vision, abdominal pain, vaginal bleeding, severe headache, and high fever). Proximity to a health center was grouped on the basis of median < 3 km and ≥3 km, whereas marriage duration was categorized into quartiles. This method provided smaller SE for more precise effect estimations. Moreover, descriptive statistics were used to summarize the data. The categorical variables, such as educational level, occupation, parity, household income, and media exposure, were summarized in terms of frequencies and percentages. For continuous variables such as age, data were summarized using the mean and standard deviation (SD). Chi-square tests were used to examine the associations between categorical variables and the initiation of ANC. Furthermore, binary logistic regression was used to highlight the factors associated with the timely initiation of ANC in model 1. We performed sensitivity analyses with two additional models: model 2 included all predictors except the husband’s educational level, while model 3 involved all predictors except the wife’s educational level. We considered *p*-values of less than 0.05 to be significant. The data were analyzed using SPSS version 23 (IBM Corp., Armonk, NY, USA). A fully reproducible copy of the data available in Appendix A.

## 3. Results

Table 1 shows the sociodemographic characteristics of the pregnant women included in the study. From the 395 pregnant women who were invited to participate in this study during their routine ANC visit, only 26 pregnant women refused the invitation, with a participation rate of 93.4%. The mean age of women was 30.4 years, and that of their husbands was 34.9 years. Approximately 60.0% of the husbands had less than a university education. Approximately one-third of the wives are employed, compared with 87.5% of the husbands. More than 60.0% of the women studied had multiple children. Over 80% of these women are exposed to media, have received ANC at a nearby health center, and have consistently followed up during their pregnancy. Seventy-eight percent initiated ANC in the first trimester.

Table 2 shows significant associations between proximity to healthcare facilities, media exposure, reason for ANC visits, and ANC initiation. Women living closer to healthcare facilities (within 3 km) had significantly higher rates of first-trimester initiation (85.0%) than those living farther away (72.2%). Similarly, women who were exposed to media were more likely to initiate ANC in the first trimester (80.8%) than those who were not (63.4%). Women seeking ANC for periodic follow-up had much higher rates of first-trimester initiation (85.5%) than those with specific health concerns such as swelling (43.5%) or other medical issues (32.1%).

Table 3 illustrates the associated predictors of ANC initiation within the first trimester. Wives with a university degree or higher were positively associated with early initiation of ANC, with an odds ratio of 2.369 and confidence intervals of 1.154–4.901. Conversely, husbands with a university degree or higher were negatively associated with early initiation, with a *p*-value of 0.003. With respect to the reasons for ANC visits, those with reasons other than periodic follow-up were negatively associated with earlier commencement of ANC at *p* < 0.001.

Table 4 demonstrates significantly associated predictors of educational levels with ANC initiation during the first trimester. When examined independently, higher maternal education showed a positive but non-significant association with the timely initiation of ANC. This was in contrast to the husband’s educational level, where a higher level remained significantly and negatively associated with appropriate ANC initiation, with an odds ratio of 0.405 and confidence intervals of 0.208–0.787.

Considering ANC services, the majority of the participants underwent clinical examination, their blood pressure and weight were measured, and blood tests were performed. They also received health education and medications, including vitamins. Notably, only 37.1% had an ultrasound test, which may indicate system-related limitations such as inadequate equipment, staff shortages, or referral requirements within primary healthcare centers. Additionally, approximately 50% of the patients received prophylactic vaccinations. Figure 2 shows services received by pregnant women during their ANC visits.

## 4. Discussion

This study provides valuable insights into ANC initiation patterns in the Jazan region of Saudi Arabia among pregnant women who attended for their ANC. Our findings indicate that 78.9% of pregnant women initiated ANC in the first trimester compared to the global average of 58.6% reported by Moller et al. [8]. This rate of early initiation compared with global statistics may reflect Saudi Arabia’s investment in maternal healthcare services and the implementation of programs such as the Maternal and Child Health Passport [19], which is similar to successful initiatives such as Afghanistan’s maternal and child health handbook, which has shown high retention and usage rates [28]. Local structural and sociodemographic factors are likely to influence these patterns. However, 21.1% of women still delay their ANC until the second or third trimester, missing crucial early interventions, which is a similar finding to that obtained in a study conducted in Unizah city [29].

### 4.1. Predisposing Factors

Predisposing factors, such as maternal and paternal education, influence health-seeking behavior [22]. In our study, both maternal and paternal education showed significant opposing associations with timely ANC initiation. Higher maternal education was positively associated with earlier ANC initiation, consistent with previous studies that have demonstrated the influence of education on maternal healthcare-seeking behavior [30,31]. This is because educated women may have better health literacy, greater awareness of ANC benefits, and more autonomy in decision-making regarding their healthcare. Interestingly, a higher level of husband’s education was negatively associated with early ANC initiation. Sensitivity analyses demonstrated that paternal education remained significant regardless of maternal education, whereas maternal education lost its significance when paternal education was omitted. This indicates that higher paternal education exerts a more robust influence on timely ANC initiation. From a statistical perspective, we addressed multicollinearity concerns between the two variables by calculating the variance inflation factor, which was 1.228 for paternal education and 1.443 for maternal education. Therefore, these contrasting variables represent distinct parental influences on ANC. This unexpected finding warrants further investigation, as it contradicts other studies that have shown positive associations between paternal education and maternal healthcare utilization [32]. Cultural factors and decision-making dynamics within families may influence this relationship. However, this area warrants further investigation in Saudi Arabia in terms of ANC. A study revealed that 61.4% of pregnant women were accompanied by their husbands during their antenatal visits [33]. In the context of Saudi Arabia, where it is very common to marry after establishing employment, it is possible that the professional commitments of husbands contribute to delayed initiation of ANC. Additionally, Saudis in general—and particularly those with higher education—tend to exhibit an optimistic outlook, which may foster a perception of low risk and further delay early care-seeking. In contrast, a review from Sub-Saharan Africa revealed that husbands were reluctant to accompany their wives to maternal and child health services because of the feeling of being dominated by their wives or due to their positive beliefs in traditional customs during pregnancy instead of healthcare institutions [34].

### 4.2. Enabling Factors

Enabling resources, such as household income and couples’ employment, determine access to healthcare services [22]. In our study, geographical accessibility emerged as a significant barrier to ANC initiation. Women living more than 3 km from health facilities were less likely to initiate ANC in the first trimester. This finding aligns with Andersen’s model, which identifies enabling factors such as proximity to healthcare facilities as important determinants of healthcare utilization [22]. It also corroborates findings from other studies in different settings that have identified distance as a barrier to maternal healthcare utilization [35,36]. Pugliese-Garcia et al. (2019) specifically highlighted distance as a critical barrier to maternal healthcare access in Egypt, a neighboring country to Saudi Arabia [37]. Additionally, media exposure, which underscores the importance of communication strategies, was significantly associated with ANC initiation, but when adjusted, it appeared as a non-significant predictor toward early initiation. Collectively, these findings still confirm Andersen’s model, which emphasizes enabling resources as critical determinants of healthcare use.

### 4.3. Need Factors

Women seeking care for specific health concerns rather than routine follow-up were significantly less likely to initiate ANC early. This suggests that symptom-driven healthcare-seeking behavior may lead to delayed ANC initiation, highlighting the importance of raising awareness about preventive, rather than curative, approaches to maternal healthcare.

Among the identified barriers to ANC initiation was the self-perception of being healthy. These findings align with the enabling and need components of Andersen’s model [22], where contextual enabling factors such as perceived need for healthcare influence utilization patterns. From another perspective, this finding aligns with the Health Belief Model, which suggests that perceived susceptibility and severity determine healthcare-seeking behavior [38]. However, those who feel healthy should initiate ANC early since it is a proactive approach, and various pregnancy-related conditions are asymptomatic during the first stages, such as diabetes.

### 4.4. External Environmental Factors

External environmental factors, such as healthcare infrastructure and system capacity, also influence service utilization [22]. With respect to ANC, most women received essential services such as clinical examinations, blood pressure and weight measurements, blood tests, and health education. However, only 37.1% had ultrasound tests, and approximately 50% received prophylactic vaccines. This indicates potential gaps in comprehensive ANC service provision that need to be addressed. In particular, ultrasound should be performed at least once during the first trimester, as recommended by the WHO and the Saudi Ministry of Health [1,39]. However, ultrasound utilization in our study was unexpectedly much lower (37.1%) compared with near-universal coverage reported in other regions such as Riyadh (~100%) [40,41]. Despite structural indicators, the Jazan region faces notable challenges in healthcare resources. For example, with a population of approximately 1.6 million, the region has 22 hospitals and 170 primary healthcare centers, yielding a ratio of 0.14 hospitals and 1.1 primary healthcare centers per 10,000 population. In terms of human resources, only 68 family physicians serve the region (0.4 per 10,000 population) [23,24]. Furthermore, a report by the Saudi Ministry of Health recorded the lowest adequacy rating of medical equipment in Jazan (77.9%) compared with the nearest rating of (94.2%) in the Madinah region [42]. Although these figures are general, they still highlight underlying structural gaps in the Jazan health sector, which may help explain the low utilization of ultrasound services among pregnant women. This variation draws attention to addressing this healthcare disparity, especially in terms of the regional profile of adverse birth outcomes [21], thus ensuring equitable access to essential ANC services and optimal health attainment.

### 4.5. Recommendations

By interpreting our findings through Andersen’s model [22], it is evident that ANC initiation in Jazan is influenced by an interplay of predisposing factors (education), enabling resources such as distance to healthcare centers, and need factors, which reflect the perceived health status and symptom-driven care. Additionally, even though utilization of ANC beyond initiation was not investigated in this study, external environmental conditions such as structural health system constraints can significantly influence healthcare-seeking behavior. As a result, addressing these barriers requires integrated interventions that combine health education, accessibility improvements, and system-level investments.

Digitizing the Maternal and Child Health Passport by incorporating an integrated interface through the Sehhaty platform—a platform that enables users to access health information and obtain various health services provided by different entities in the Saudi Arabian healthcare sector [43]—can be an efficient way to provide a tailored pregnancy journey for pregnant women, as it can enhance service usage [44]. In simple terms, automated reminders for visits, supplements, and alert systems based on personalized data entry, as well as targeted educational interventions tailored to specific contexts, can be provided. Digital health interventions, as reviewed by Knop et al. (2024), have also been shown to improve maternal care in low- and middle-income countries [45]. It could also benefit the National Platform for Health and Insurance Exchange Services “nphies” platform by enhancing the continuity of care across multiple providers [46], serving as a portable pregnancy history.

### 4.6. Limitations

This study has several limitations. First, the cross-sectional study design allows for the capture of a snapshot at a specific point in time, which was during the third trimester of pregnancy in this study, precluding the assessment of ANC utilization and participant attrition throughout pregnancy. Second, pregnant individuals who had pregnancy complications, comorbidities, or psychological problems or who refused to participate were excluded from the study, which may introduce selection bias, and our findings are likely to reflect routine ANC journeys instead of specific pathways in healthcare. Therefore, future studies should include high-risk pregnancies. Third, the sampling strategy does not support the findings’ generalizability to pregnant women outside of our study due to clinic-based data collection. Fourthly, several variables were collected from pregnant women and are subject to recall bias. Finally, the quality of the provided care was not measured in a numerical way, e.g., via a Likert scale, which would provide insights into areas that need improvement.

## 5. Conclusions

While ANC initiation in the Jazan region shows promising trends, significant barriers to optimal maternal healthcare initiation remain. Geographical accessibility in terms of distance >3 km, higher educational levels for husbands, and need factors are key factors affecting ANC early initiation patterns. Targeted interventions to overcome these geographical barriers and the promotion of ANC as a proactive, preventive practice—rather than a service sought only when problems arise—could substantially improve maternal and child health outcomes in the region. It is strongly recommended that future research explore the unexpected negative association between husband’s education and early ANC initiation and that qualitative investigations be conducted into family dynamics and decision-making processes regarding maternal healthcare utilization in this cultural context within families. Additionally, low utilization of ultrasound tests in the region should be investigated.

## Figures and Tables

**Figure 1 healthcare-13-02449-f001:**
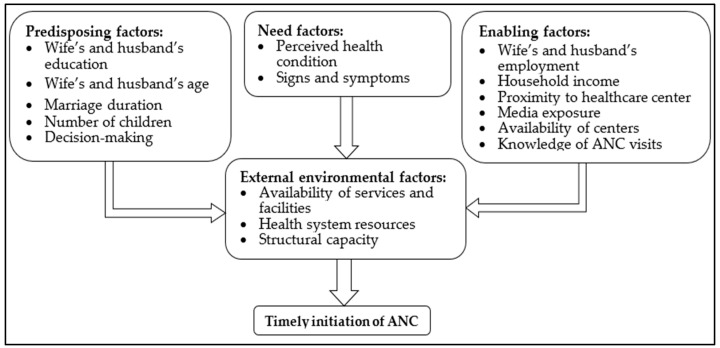
Andersen’s behavioral model framework applied to timely ANC initiation.

**Figure 2 healthcare-13-02449-f002:**
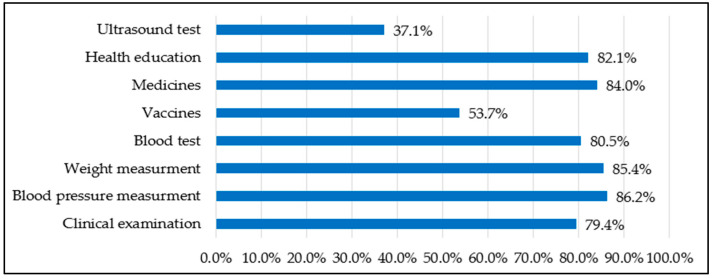
Basic ANC services have been received by pregnant women.

**Table 1 healthcare-13-02449-t001:** Sociodemographic characteristics of participants.

Variables		n (%)/Mean (SD)
Wife’s age in years		30.4 (6.6)
Husband’s age in years		34.9 (9.8)
Wife’s educational level	Below university degree	192 (52.0)
	University degree or above	177 (48.0)
Husband’s educational level	Below university degree	223 (60.4)
	University degree or above	146 (39.6)
Wife’s employment	Housewife	249 (67.5)
	Employed	120 (32.5)
Husband’s employment	Not working	46 (12.5)
	Employed	323 (87.5)
Parity	Null or primiparous	134 (36.3)
	Multiparous	235 (63.7)
Household income in SR	<5000	82 (22.2)
	5000–9999	207 (56.1)
	≥10,000	80 (21.7)
Proximity in km	≤3	193 (52.3)
	>3	176 (47.7)
Media exposure	Yes	328 (88.9)
	No	41 (11.1)
Knowledge of recommended ANC visits	Correct answer	134 (36.3)
	Wrong answer	235 (63.7)
Where did you get ANC?	Nearby healthcare center	303 (82.1)
	Government hospital	27 (7.3)
	Private hospital	39 (10.6)
Reason for ANC visit	Periodic follow-up	318 (86.2)
	Swelling of hands/face	23 (6.2)
	Others	28 (7.6)
Marriage duration	≤3 years	77 (20.9)
	4–7 years	94 (25.5)
	8–11 years	110 (29.8)
	≥12 years	88 (23.8)
ANC initiation	First trimester	291 (78.9)
	Second or third trimesters	78 (21.1)

**Table 2 healthcare-13-02449-t002:** Distribution of ANC initiation by sociodemographic factors.

Variables		ANC Initiation		Total	*p*-Value
		First Trimester	Second or Third Trimester		
		n (%)	n (%)		
Total		291 (78.9)	78 (21.1)	369	
Wife’s age in years	<20	25 (86.2)	4 (13.8)	29	0.812
	20–24	38 (74.5)	13 (25.5)	51	
	25–29	67 (77.0)	20 (23.0)	87	
	30–34	78 (80.4)	19 (19.6)	97	
	35–39	58 (77.3)	17 (22.7)	75	
	≥40	25 (83.3)	5 (16.7)	30	
Husband’s age in years	<20	10 (83.3)	2 (16.7)	12	0.437
	20–24	21 (77.8)	6 (22.2)	27	
	25–29	43 (71.7)	17 (28.3)	60	
	30–34	66 (81.5)	15 (18.5)	81	
	35–39	70 (75.3)	23 (24.7)	93	
	≥40	81 (84.4)	15 (15.6)	96	
Wife’s educational level	Below university degree	146 (76.0)	46 (24.0)	192	0.167
	University degree or above	145 (81.9)	32 (18.1)	177	
Husband’s educational level	Below university degree	183 (82.1)	40 (17.9)	223	0.063
	University degree or above	108 (74.0)	38 (26.0)	146	
Wife’s employment	Housewife	197 (79.1)	52 (20.9)	249	0.863
	Employed	94 (78.3)	26 (21.7)	120	
Husband’s employment	Not working	36 (78.3)	10 (21.7)	46	0.915
	Employed	255 (78.9)	68 (21.1)	323	
Parity	Null or primiparous	110 (82.1)	24 (17.9)	134	0.251
	Multiparous	181 (77.0)	54 (23.0)	235	
Household income in SR	<5000	62 (75.6)	20 (24.4)	82	0.425
	5000–9999	162 (78.3)	45 (21.7)	207	
	≥10,000	67 (83.8)	13 (16.3)	80	
Proximity in km	≤3	164 (85.0)	29 (15.0)	193	0.003
	>3	127 (72.2)	49 (27.8)	176	
Media exposure	Yes	265 (80.8)	63 (19.2)	328	0.015
	No	26 (63.4)	15 (36.6)	41	
Knowledge of recommended ANC visits	Correct answer	100 (74.6)	34 (25.4)	134	0.132
	Wrong answer	191 (81.3)	44 (18.7)	235	
Where did you get ANC?	Nearby healthcare center	243 (80.2)	60 (19.8)	303	0.109
	Government hospital	17 (63.0)	10 (37.0)	27	
	Private hospital	31 (79.5)	8 (20.5)	39	
Reason for ANC visit	Periodic follow-up	272 (85.5)	46 (14.5)	318	<0.001
	Swelling of hands/face	10 (43.5)	13 (56.5)	23	
	Others	9 (32.1)	19 (67.9)	28	
Marriage duration	≤3 years	64 (83.1)	13 (16.9)	77	0.053
	4–7 years	65 (69.1)	29 (30.9)	94	
	8–11 years	88 (80.0)	22 (20.0)	110	
	≥12 years	74 (84.1)	14 (15.9)	88	

**Table 3 healthcare-13-02449-t003:** Factors associated with ANC initiation within the first trimester (Model 1).

Predictive Variables		ANC Initiation: First Trimester			
		β	SE	aOR (95% CI)	*p*-Value
Age in years	Wife’s age	−0.038	0.037	0.963 (0.897–1.035)	0.303
	Husband’s age	0.016	0.021	1.016 (0.976–1.058)	0.432
Wife’s educational level *	Below university degree	Reference			
	University degree and above	0.863	0.371	2.369 (1.154–4.901)	0.020
Husband’s educational level *	Below university degree	Reference			
	University degree and above	−1.051	0.353	0.350 (0.175–0.699)	0.003
Wife’s employment	Housewife	Reference			
	Employed	0.174	0.394	1.190 (0.550–2.574)	0.659
Husband’s employment	Not working	Reference			
	Employed	0.652	0.562	1.919 (0.638–5.775)	0.264
Parity	Null or primiparous	0.406	0.448	1.501 (0.624–3.607)	0.364
	Multiparous	Reference			
Household income in SR	<5000	−0.295	0.597	0.744 (0.231–2.397)	0.621
	5000–9999	−0.136	0.460	0.873 (0.354–2.151)	0.767
	≥10,000	Reference			
Proximity in km	≤3	Reference			
	>3	−0.673	0.313	0.510 (0.276–0.941)	0.031
Media exposure	Yes	Reference			
	No	−0.638	0.468	0.528 (0.211–1.321)	0.172
Knowledge of recommended ANC visits	Correct answer	Reference			
	Wrong answer	0.067	0.329	1.069 (0.561–2.038)	0.839
Where did you get your ANC?	Nearby healthcare center	Reference			
	Government hospital	−0.678	0.608	0.508 (0.154–1.671)	0.265
	Private hospital	−0.265	0.483	0.768 (0.298–1.978)	0.584
Reason for ANC visit	Periodic follow-up visit	Reference			
	Swelling of hands	−1.899	0.523	0.150 (0.054–0.417)	<0.001
	Others	−2.399	0.518	0.091 (0.033–0.251)	<0.001
Marriage duration in years	≤3 years	−0.163	0.711	0.849 (0.211–3.422)	0.818
	4–7 years	−0.926	0.549	0.396 (0.135–1.163)	0.092
	8–11 years	−0.222	0.455	0.801 (0.328–1.955)	0.625
	≥12 years	Reference			
Reported barriers	Lack of services/facilities	−0.547	0.381	0.578 (0.274–1.221)	0.151
	Lack of physician recommendation	0.433	0.484	1.542 (0.598–3.979)	0.370
	Feeling healthy	−0.753	0.369	0.471 (0.229-0.970)	0.041

* Collinearity diagnostics (variance inflation factors) were 1.443 for wife’s education and 1.228 for husband’s education.

**Table 4 healthcare-13-02449-t004:** Sensitivity analyses of the significant educational level predictors associated with ANC initiation within the first trimester.

Predictive Variables		ANC Initiation: First Trimester			
		β	SE	aOR (95% CI)	*p*-Value
Model 2	Wife’s educational level				
	Below university degree	Reference			
	University degree and above	0.613	0.352	1.847 (0.926–3.682)	0.081
Model 3	Husband’s educational level				
	Below university degree	Reference			
	University degree and above	−0.905	0.339	0.405 (0.208–0.787)	0.008

## Data Availability

The original contributions presented in this study are included in the article/Appendix A. Further inquiries can be directed to the corresponding author.

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
