# Peer review of "Initiation of Antenatal Care Among Pregnant Women in Saudi Arabia: An Application of Andersen’s Behavioral Model Using a Cross-Sectional Study"

_healthcare, 2025, doi:10.3390/healthcare13192449_

Round 1

Reviewer 1 Report

Comments and Suggestions for Authors

The manuscript is written well, and all the sections of the manuscript are well-aligned. However, the manuscript has some methodological, analytical, and interpretive concerns that need attention before it can be considered for publication. 

Introduction

  • Minor edit: Use “Andersen’s Behavioral Model” (not Anderson’s).
  • The authors have set a good layout for the background of the study. However, some statistics are overly focused on the global context and could be shortened to focus on the Saudi/Jazan context.
  • The authors should better explain why Jazan specifically has higher adverse outcomes (Lines 101–104).
  • The application of Andersen’s model is justified, but one should note how it differs from earlier Saudi studies (e.g., Alanazy & Brown, 2020).

Methods

  • Lines 131–141: Excluding women with pregnancy complications may bias findings. This limitation should be acknowledged earlier, or discuss why you excluded this population.
  • Data Collection: Questionnaire details are adequate, but verbal informed consent (Line 159) might raise ethical concerns; justify why written consent was not possible.
  •  Analysis): Clear description. However, collapsing categories (e.g., income, proximity, education) may reduce nuance. Justify.
  • Study framework: The author introduced Anderson’s model at various places, such as the introduction, methods, and contextualized the findings in the discussion section. I would appreciate it if the author could also put a figure of this model. This is not highly recommended, but the readers can easily understand the visual presentation of this model.

Results

  • Table 2: I would suggest adding p-values in text to highlight the strength of associations.
  • Regression: The Main findings are interesting. The paternal education effect (negative) requires cautious reporting. Could it be due to collinearity with maternal education or socio-cultural decision-making? A sensitivity analysis might strengthen the argument.
  • Line 249–254: The ultrasound gap (37.1%) is striking. Needs stronger link with possible system constraints (equipment, staff availability, referral protocols).

Discussion

  • Lines 257–268: Good comparison with global data. However, overemphasis on “higher than global average” should be toned down, as local context differs.
  • Line 275–283: Paternal education finding is highlighted, but the speculative explanation is thin. Authors should explore gender dynamics, health decision-making roles, or possibly reverse causality.
  • Lines 297–302: Self-perceived health as a barrier is valid. The author may link or reference the health belief models for stronger theoretical grounding and foundation.
  • Limitations: The limitation section is written well and comprehensive, but recall bias and generalizability issues could be emphasized in this section.
  • Lines 343–346 are the qualitative follow-up that should be stressed more strongly to understand and explore the context of the findings.

Author Response

Response Letter

Manuscript ID: healthcare-3788175

Previous title: Initiation of Antenatal Care among Pregnant Women in Saudi Arabia: an Application of Anderson’s Behavioral Model Using a Cross Sectional Study

New title: Initiation of Antenatal Care among Pregnant Women in Saudi Arabia: An Application of Andersen’s Behavioral Model Using a Cross Sectional Study (Minor edits: (An) is capitalized, and Andersen’s instead of Anderson’s)

First of all, we would like to thank the editor for giving us the opportunity to revise our manuscript, and also we extend our pleasure to the reviewers for their valuable time and comments. Additionally, we will try our best to address points one-by-one. Below, we describe how we handled these comments regarding our previous submission. The changes are highlighted in the manuscript. And the listed page and line numbers in this letter are also highlighted and correspond to the manuscript’s highlighted content.

Reviewer 1 comments and suggestions

  • The manuscript is written well, and all the sections of the manuscript are well-aligned. However, the manuscript has some methodological, analytical, and interpretive concerns that need attention before it can be considered for publication.

Response:  Thank you for such recognition. More importantly, we are willing to carefully consider these comments one-by-one which were purposively-generated during this insightful peer review process.

Introduction

  • Minor edit: Use “Andersen’s Behavioral Model” (not Anderson’s).

Response: We have amended the name throughout the whole manuscript.

  • The authors have set a good layout for the background of the study. However, some statistics are overly focused on the global context and could be shortened to focus on the Saudi/Jazan context.

Response: We have added valuable details pertaining to Saudi Arabia as well as Jazan region. These details included reported determinants of timely ANC initiation in addition to the proportion of women with adverse birth outcomes and its prevalence. Please see page: 3; Lines: 103-106 and 108-111.

  • The authors should better explain why Jazan specifically has higher adverse outcomes (Lines 101–104).

Response: We have added supporting evidence that may explain higher adverse outcomes. Kindly see page: 3; Lines: 111-114. Additionally, we have linked adverse outcomes disparities to our findings of low ultrasound test utilization, as a possible contributing factor. Page: 12; Lines: 380-383.

  • The application of Andersen’s model is justified, but one should note how it differs from earlier Saudi studies (e.g., Alanazy & Brown, 2020).

Response: We have justified its difference from other studies, particularly in terms of allowing us to assess initiation from a comprehensive view, where other studies either did not account for these factors or covered them partially. Please see page: 3; Lines 119-124.

Methods

  • Lines 131–141: Excluding women with pregnancy complications may bias findings. This limitation should be acknowledged earlier, or discuss why you excluded this population.

Response: We excluded them because we want to capture the routine situation, not the complicated situations as ANC initiation time will be biased because of these medical conditions can be characterized by different care pathways, thus, different services. Additionally, care may be initiated just for these complication instead of seeking purely ANC. Please note page: 4; Lines: 150-152.

  • Data Collection: Questionnaire details are adequate, but verbal informed consent (Line 159) might raise ethical concerns; justify why written consent was not possible.

Response: Written consent was not pursued because of cultural sensitivities and varying literacy levels among participants, which could have discouraged participation. Instead, verbal consent was used to ensure inclusivity and comfort, and this procedure was explicitly reviewed and approved by the Jazan Health Ethics Committee (protocol No. 22102, 27 September 2022), in line with the Declaration of Helsinki. Please see page: 5; Lines: 177-179.

  • Analysis): Clear description. However, collapsing categories (e.g., income, proximity, education) may reduce nuance. Justify.

Response: In the preliminary analyses, we noticed larger sampling errors associated with very low frequencies in some categories. We tried to minimize these errors by collapsing categories, therefore, standard/sampling errors were reduced as a result of increasing sample size in these categories. Purposively, we reported SE in the regression analysis so that readership can see how they are substantially below 1. We justified it in the manuscript too. Kindly see page: 5; Lines: 218-219 and page: 6; Lines: 224-225.

  • Study framework: The author introduced Anderson’s model at various places, such as the introduction, methods, and contextualized the findings in the discussion section. I would appreciate it if the author could also put a figure of this model. This is not highly recommended, but the readers can easily understand the visual presentation of this model.

Response: We appreciate your concern for readers easy comprehension. In fact, we have structured a figure containing Anderen’s model in its various domains, including the external environmental factors. Kindly look at page 4.

Results

  • Table 2: I would suggest adding p-values in text to highlight the strength of associations.

Response: All p-values were added. Kindly look at Table 2; Pages: 7-8.

  • Regression: The Main findings are interesting. The paternal education effect (negative) requires cautious reporting. Could it be due to collinearity with maternal education or socio-cultural decision-making? A sensitivity analysis might strengthen the argument.

Response: We greatly appreciate such comment. Indeed, we have conducted sensitivity analyses including two additional models. These models were clarified in the data management and analysis subsection in the materials and methods, and in results were interpreted cautiously. Please see page: 6; Lines: 231-234 and page: 9; Lines: 269-276. Moreover, we provided variance inflation factors for the two variables, kindly look at the end of table 3 at page 9. After sensitivity analyses, both predictors have same directions but only husband education remained significant, and given the variance inflation factors, it seems to be below the threshold of multicollinearity concerns. As a result, in the discussion section, we have rephrased the relevant sections, reflecting findings after the sensitivity analysis. Kindly see page: 11; Lines: 302-305 and also page: 11; Lines: 308-316. We also discussed the possible role of familial decision making in Saudi Arabia and recommended future investigation, please see page: 11; Lines: 316-321.

Accordingly, we rephrased the abstract to maintain a consistent and valid presentation of the findings throughout the manuscript. Look at pages: 1 and 2; Lines: 37-39 and 44-47. Nevertheless, sentences pertaining to maternal education in the conclusion section were omitted from the recommendations of future studies.

  • Line 249–254: The ultrasound gap (37.1%) is striking. Needs stronger link with possible system constraints (equipment, staff availability, referral protocols).

Response: We have addressed this point by adding “Notably, only (37.1%) had an ultrasound test, which may indicate system-related limitations such as inadequate equipment, staff shortages, or referral requirements within primary healthcare centers”. Page: 10; Lines: 280-283. Furthermore, this figure was discussed more in the discussion section. Please see page: 12; Lines: 365-383.

Discussion

  • Lines 257–268: Good comparison with global data. However, overemphasis on “higher than global average” should be toned down, as local context differs.

Response: We have rephrased the sentence and justified the role of structural and sociodemographic factors in influencing this pattern. Please read Page: 10; Lines: 289-297.

  • Line 275–283: Paternal education finding is highlighted, but the speculative explanation is thin. Authors should explore gender dynamics, health decision-making roles, or possibly reverse causality.

Response: We have provided possible explanations considering available data and local- cultural context. Page: 11; Lines: 321-326. These speculations are additional to other reasons mentioned, such as in Sub-Saharan Africa.

  • Lines 297–302: Self-perceived health as a barrier is valid. The author may link or reference the health belief models for stronger theoretical grounding and foundation.

Response: Additional to Andersen’s Model, we have introduced the Health Belief Model from a perspective of susceptibility and severity. Kindly see page: 11; Lines: 355-357.

  • Limitations: The limitation section is written well and comprehensive, but recall bias and generalizability issues could be emphasized in this section.

Response: Such important considerations are justified within the limitation section. Page 13; Lines: 414-417.

  • Lines 343–346 are the qualitative follow-up that should be stressed more strongly to understand and explore the context of the findings.

Response: We have strongly recommended the investigation of our study major findings. Page: 13; Lines: 428-433.

Reviewer 2 Report

Comments and Suggestions for Authors

The authors examine the utilization of prenatal services among women resident in a specific region of Saudi Arabia.  This cross-sectional study uses the Anderson behavior model as an organizing framework.  The authors provide an adequate justification for the study in the Introduction.  Their description of the Methods is generally complete, but needs to be expanded.  They present a straight forward statement of the Results, with appropriate tables and figures.  The Discussion integrates the results into the larger literature, but could be further developed.

The authors could improve their paper by addressing several points.

The most important issue that the authors do not address is the problem faced by all research that uses a clinic-based sample – the characteristics of those who do not attend any clinic for prenatal care and the factors associated with their not utilizing any prenatal services.  The sample for this study is limited to those who attended a clinic.  We do not know anything about those women with no prenatal care.  The authors should address this point in the presentation of Methods and in their Discussion.

The authors excluded women who “had pregnancy complications or comorbidities.”  They list this to be a limitation.  They do not justify this exclusion.  These women are important for understanding in the causes and content of prenatal services.  The type of data collection – an interview – would not increase the risk experienced by these women.  This exclusion must be justified. 

The authors excluded women who “refused to participate.”  They list this to be a limitation.  This is appropriate.  However, the authors provide no information on the number of refusals beyond noting in Table A1 that “reasons for non-participation at each stage” were not applicable.

The authors use the Anderson model to structure their research.  However, they do not use the Anderson model to structure their Discussion.  Rather, it appears that two references to the Anderson model are asides.  The Discussion should be restructured and expanded to reflect the Anderson model.  This revision would made the research useful beyond a case report for Saudi Arabia.

Author Response

Response Letter

Manuscript ID: healthcare-3788175

Previous title: Initiation of Antenatal Care among Pregnant Women in Saudi Arabia: an Application of Anderson’s Behavioral Model Using a Cross Sectional Study

New title: Initiation of Antenatal Care among Pregnant Women in Saudi Arabia: An Application of Andersen’s Behavioral Model Using a Cross Sectional Study (Minor edits: (An) is capitalized, and Andersen’s instead of Anderson’s)

First of all, we would like to thank the editor for giving us the opportunity to revise our manuscript, and also we extend our pleasure to the reviewers for their valuable time and comments. Additionally, we will try our best to address points one-by-one. Below, we describe how we handled these comments regarding our previous submission. The changes are highlighted in the manuscript. And the listed page and line numbers in this letter are also highlighted and correspond to the manuscript’s highlighted content.

Reviewer 2 comments and suggestions

  • The authors examine the utilization of prenatal services among women resident in a specific region of Saudi Arabia.  This cross-sectional study uses the Anderson behavior model as an organizing framework.  The authors provide an adequate justification for the study in the Introduction.  Their description of the Methods is generally complete, but needs to be expanded.  They present a straight forward statement of the Results, with appropriate tables and figures.  The Discussion integrates the results into the larger literature, but could be further developed.

Response: We have added specific details in the materials and methods sections. These details included an explanation of excluding complicated and comorbid pregnant women, justification of verbal consent, collapsing certain variables, and in addition to providing a figure and performing additional sensitivity analysis of two different models to eliminate the role of collinearity between maternal and paternal educational levels. All of these details are specified down.

  • The authors could improve their paper by addressing several points.

Response: We are willing to comply with any suggestions/comments to improving our manuscript.

  • The most important issue that the authors do not address is the problem faced by all research that uses a clinic-based sample – the characteristics of those who do not attend any clinic for prenatal care and the factors associated with their not utilizing any prenatal services.  The sample for this study is limited to those who attended a clinic.  We do not know anything about those women with no prenatal care.  The authors should address this point in the presentation of Methods and in their Discussion.

Response: We have addressed this point in the methods section, kindly see page: 4; Lines: 152-154. And early in the discussion section, page 10; Lines: 289-290. Additionally, in the limitation subsection of the discussion, we acknowledged our sample and its potential role in introducing selection bias, page 13, line 412.

  • The authors excluded women who “had pregnancy complications or comorbidities.”  They list this to be a limitation.  They do not justify this exclusion.  These women are important for understanding in the causes and content of prenatal services.  The type of data collection – an interview – would not increase the risk experienced by these women.  This exclusion must be justified. 

Response: We have justified this exclusion in the method section. Kindly see page 4; Lines: 150-152. Accordingly, in the limitation section as well, we acknowledged this exclusion as a limitation, and introduced its role in potential selection bias. More importantly, we recommended the inclusion of high-risk pregnancies in future studies. Please see Page: 13; Lines: 412-414.

  • The authors excluded women who “refused to participate.”  They list this to be a limitation.  This is appropriate.  However, the authors provide no information on the number of refusals beyond noting in Table A1 that “reasons for non-participation at each stage” were not applicable.

Response: Thank you for this comment. Indeed, data collectors did not handle the data analysis, and the designated author for the data analysis did not verify this information. However, after verification, there was a total of 26 pregnant women who refused to participate in our study. We have documented this information early in the result section. Kindly look at Table A1 and page: 6; Lines: 239-241.

  • The authors use the Anderson model to structure their research.  However, they do not use the Anderson model to structure their Discussion.  Rather, it appears that two references to the Anderson model are asides.  The Discussion should be restructured and expanded to reflect the Anderson model.  This revision would made the research useful beyond a case report for Saudi Arabia.

Response: We have restructured the discussion according to the Andersen’s model thoroughly. Kindly look at page 4, we have provided a conceptual figure containing the model factors. Additionally, we have guided our discussion using subheadings such as predisposing factors, enabling factors, needs, external factors etc. Please see pages 10-13.
